# Role of Nutrition in the Management of Patients with Multiple Endocrine Neoplasia Type 1

**DOI:** 10.3390/nu16111576

**Published:** 2024-05-23

**Authors:** Monica Marinari, Francesca Marini, Francesca Giusti, Maria Luisa Brandi

**Affiliations:** 1Department of Pharmacy, University of Pisa, 56120 Pisa, Italy; monicamarinari83@gmail.com; 2Fondazione Italiana Ricerca Sulle Malattie dell’Osso (FIRMO Onlus), 50129 Florence, Italy; francesca.marini@unifi.it; 3Department of Experimental and Clinical Biomedical Sciences, University of Florence, 50139 Florence, Italy; francesca.giusti@unifi.it; 4Donatello Bone Clinic, Villa Donatello Hospital, 50019 Sesto Fiorentino, Italy

**Keywords:** MEN1, multiple endocrine neoplasia type 1 (MEN1), neuroendocrine tumors, pancreatic neuroendocrine tumors, nutrition, dietary habits, quality of life

## Abstract

Multiple endocrine neoplasia type 1 (MEN1) is a rare syndrome caused by inactivating mutations in the MEN1 tumor suppressor gene. The three main clinical manifestations of MEN1 are primary hyperparathyroidism (PHPT), duodenal–pancreatic neuroendocrine tumors (DP-NETs) and anterior pituitary tumors. Endocrine tumors in patients with MEN1 differ from sporadic tumors because of their younger age at onset, common multiple presentations and the different clinical course. MEN1 is characterized by a complex clinical phenotype; thus, patients should be followed by a multidisciplinary team of experts that includes an endocrinologist, a surgeon, a oncologist, a radiotherapist, and not least, a nutritionist. It is important to remember the fundamental role that diet plays as a primary prevention tool, together with a healthy and active lifestyle in preventing osteoporosis/osteopenia and reducing the risk of developing kidney stones due to hypercalciuria, two frequent clinical complications in MEN1 patients. Is very important for MEN1 patients to have an adequate intake of calcium, vitamin D, magnesium and phosphate to maintain good bone health. The intake of foods containing oxalates must also be kept under control because in combination with calcium they concur to form calcium oxalate crystals, increasing the risk of nephrolithiasis. Another aspect to consider is the management of patients with pancreatic neuroendocrine tumors undergoing major surgical resections of the pancreas that can lead to alterations in digestion and absorption mechanisms due to partial or total reduction in pancreatic enzymes such as amylase, lipase, and protease, resulting in malabsorption and malnutrition. Therefore, the nutritionist’s aim should be to devise a dietary plan that takes into consideration each single patient, educating them about a healthy and active lifestyle, and accompanying them through various life stages by implementing strategies that can enhance their quality of life.

## 1. Introduction

Multiple endocrine neoplasia type 1 (MEN1) is a rare congenital tumor syndrome characterized by the development of multiple neuroendocrine tumors in different glands, in a single patient during the lifetime. It is caused by an inactivating heterozygous germline mutation of the MEN1 tumor suppressor gene, and somatic loss/inactivation of the second copy of the gene, leading to loss of oncosuppressor function of the menin protein, which causes the development of hyperplasia or tumors in specific neuroendocrine tissues [1,2,3,4,5,6,7].

The three main clinical manifestations of MEN1 are

Hyperplasia or adenomas of the parathyroid glands in over 95% of cases, causing primary hyperparathyroidism (PHPT), characterized by excessive secretion of parathyroid hormone (PTH). It is generally less severe than its sporadic counterpart, with an average age of onset around 20–25 years, and can remain normocalcemic and asymptomatic even for several years from the onset of the parathyroid alteration [8].Functioning and non-functioning gastro–entero–pancreatic neuroendocrine tumors (GEP-NETs) (80% of cases) [8].Anterior pituitary tumors in about 50% of cases (including both hormone-secreting or silent adenomas) [1,8].

Other cancers associated with MEN1 include thymus, lung, and gastric type 2 neuroendocrine tumors (NETs), adrenocortical tumors, pheochromocytomas, facial angiofibromas, collagenomas, hibernomas, meningiomas, ependymomas, leiomyomas, and lipomas, and there is an increased risk of developing breast cancer in female patients [2].

Over 1300 different somatic and germline mutations and 20 benign polymorphisms of the MEN1 gene have been identified but without any clear genotype–phenotype correlations. The MEN1 clinical phenotype differs among mutated members of the same family and even between homozygous twins. Therefore, it has been suggested that epigenetic mechanisms triggered by environmental factors, including diet, may influence the disease phenotype in *MEN1* mutation carriers.

Nutrition can be a valuable aid, together with a healthy and active lifestyle, in the prevention of osteoporosis, and in reducing the risk of developing kidney stones, two frequent clinical complications in patients with MEN1 syndrome and PHPT.

Another important aspect to be considered is the management of MEN1 patients with multiple GEP-NETs, who have undergone major resections of the duodenum and/or the pancreas, with consequent malabsorption and malnutrition, worsening the general clinical picture and the quality of life.

This review aims to outline nutritional indications for MEN1 patients, to direct them towards the “right diet” to follow, both as prevention and as treatment of the main nutritional deficiencies they may encounter, emphasizing the importance that MEN1 patients should be followed a multidisciplinary team that includes not only the endocrinologist, the surgeon, the oncologist, the radiotherapist, but also the nutritionist/dietitian.

## 2. Importance of Nutrition in MEN1 Patients with PHPT

Parathyroid dysfunction is the first clinical manifestation in most patients, with a mean age of occurrence at 20–25 years, and progressively increasing with age up to over 95% of cases after 50 years of age.

Even if MEN1 PHPT is usually milder than the sporadic counterpart and can have an indolent course for years, a non-adequately treated MEN1 PHPT may be associated with hypercalcemia, and lead to prolonged hypercalciuria, increased renal excretion of phosphates, and probable hypo-magnesia, due to increased renal excretion of magnesium.

PHPT-derived prolonged hypercalciuria leads to the development of kidney stones, with an increased risk of reduced proper kidney function.

Untreated PHPT also contributes to an early-onset reduction in bone mass; MEN1 patients with PHPT showed a significant reduction in bone mineral density (BMD), associated with an increased likelihood of fragility fracture, compared to the general population [9,10,11,12,13].

Moreover, it is suspected that *MEN1* mutations may directly affect the activity of bone cells [7,10,14,15].

In MEN1 patients with PHPT nutritional habits can substantially contribute to the occurrence and worsening of the two above mentioned PHPT-derived clinical conditions, early loss of bone mass and nephrolithiasis/nephrocalcinosis.

The nutritionist/dietitian must take into account these clinical aspects, in drawing up the MEN1 patient’s food plan, paying attention primarily to the amount of dairy dietary calcium intake, but also granting a correct intake of phosphorus, magnesium, and vitamin D, all factors that regulate bone metabolism. In addition, such a personalized eating plan must also consider another very important aspect for MEN1 patients, namely that of increased urinary calcium excretion. Indeed, some foods should be limited in the daily diet due to their high concentration of oxalates that can form calcium oxalate crystals at the kidney levels, further increasing the risk of kidney stones or worsening an already present state of nephrolithiasis, both in terms of number of new stones and/or enlargement of pre-existing ones. 

### 2.1. Dietary Interventions for Prevention and Management of Osteopenia/Osteoporosis in MEN1 PHPT

A state of constant PTH hypersecretion, not followed by an adequate dietary intake of calcium and proteins, can lead to progressive bone demineralization and, thus, osteoporosis within a short time. Trying to accumulate as much bone mass as possible during the first and second decades of life, during which peak bone mass is normally reached, may be a strategy to slow down the early demineralization process due to PHPT.

Many factors influence bone mass, and, although some of these, such as age, sex, and genotype, cannot be changed, others, such as diet and lifestyle (cigarette smoking, alcohol intake, and physical activity) are modifiable. Therefore, the best way to prevent bone loss in MEN1 patients is to provide an adequate intake of calcium and other nutrients, such as proteins, phosphorus, magnesium and vitamin D, starting from childhood, or as early as possible after the diagnosis of the syndrome.

#### 2.1.1. Calcium Dietary Intake

The recommended daily dietary intake of calcium ranges between 700 and 1200 mg/day, (Table 1) as stated at the Italian level by the Reference Levels of Nutrient and Energy Intake for the Italian Population (LARN) [16].

Although dairy products are the largest and best-known dietary source of calcium, we must not forget that natural mineral waters are also an important source.

Water is the main constituent of the human body and it is involved in many biological functions. According to LARN, an intake of 1.2 to 2.5 L of water per day should be ensured, because good hydration is essential to maintain body water balance, although needs may vary among people, depending on age, physical activity, and personal circumstances. The United States Department of Agriculture (USDA) reports that water intake from food, beverages, and drinking water should be 2.7 L per day in women aged 19–50 years [16].

The European 2009/54/EC directive classified natural mineral waters according to their mineral content, indicating ‘Water with Calcium’ if the calcium content is >150 mg/L. Mineral waters high in calcium could, therefore, be recommended to provide both a dietary source of calcium and adequate hydration, without caloric intake [17].

By the 1990s, some studies have been conducted to assess the bioavailability of calcium contained in calcium-rich mineral water compared to that of calcium taken in from dairy products [18,19,20]. In most studies, calcium absorption from mineral waters was measured both directly, using an isotope tracer technique, or indirectly, by evaluating serum PTH concentration, and urinary and serum calcium concentration [17]. Although the number of these studies is limited and, in general, small groups of participants have been recruited, their converging results have led to a single, concordant conclusion: the bioavailability of calcium from calcium-rich mineral waters is possibly superior to that of calcium contained in milk and dairy products [21].

Specifically, Bacciottini et al. [18] measured the bioavailability of calcium from high-calcium mineral water in 27 healthy subjects, comparing it to milk, showing that calcium from mineral water and milk are similarly bioavailable. Overlapping results were reported in a study by Heaney and Dowell, which compared the bioavailability of calcium from calcium-rich water and milk in 18 healthy women [17,20].

Following these two studies, the role of calcium-rich mineral waters on mineral and bone metabolism was evaluated, albeit in a relatively small number of subjects. Most studies have shown that calcium-rich mineral waters have a positive impact on both bone biomarkers and densitometric parameters [20,22].

Dairy products are another main dietary source of calcium, which provide high levels of this mineral, but also phosphorous and protein. One liter of cow’s milk provides 1200 mg/L of calcium, 1150 mg/L of phosphorus, 32–35 g/L of protein, such as casein and whey protein, which also contain a number of cell growth factors, along with calories, trace elements and vitamins. Whey protein is more rapidly digested and absorbed than casein. Despite variations in milk composition according to breed, season and food, commercially available milk is usually standardized and sometimes fortified with vitamin D. Depending on the species, the nutrient content can vary considerably [21].

When talking about plant-based beverages to replace cow’s milk, it is important to take into account that they can have a similar concentration of protein content, but they strictly require the addition of minerals and carbohydrates to reach calcium and caloric intakes comparable to cow’s milk. Therefore, if cow’s milk is replaced by unfortified and unsupplemented vegetable drinks, patients may incur some nutrient deficiencies. Particular attention must be paid to children and adolescents who exclusively receive such vegetable drinks, and who may experience a reduction in peak bone mass, with the consequent risk of osteopenia and osteoporosis in adulthood.

A very important aspect to take into account is the bioavailability of calcium within the food matrix. Indeed, to be absorbed, this mineral must be present in the intestine in its ionized form. The solubilization and ionization of calcium in food often occur in the acidic environment of the stomach, where the solubility of calcium salts and complexes increases. It is crucial to remember that calcium absorption is governed by several factors, such as the concentration of ionized calcium, which in turn depends on the pH of the environment. Calcium content in foods is often represented by calcium salts of phosphate, carbonate, citrate, oxalate or calcium complexes with proteins. These molecules have a significantly lower solubility than free calcium ion, and their intestinal absorbance has a strong dependence on biological environment pH. The solubility of calcium carbonate, calcium citrate and calcium phosphate progressively increase when pH decreases. The presence of phthalate and oxalate in some food groups can also severely limit calcium absorption because complexes that form are not solubilized and ionized despite the acidic pH found during gastric transit. Thus, calcium fortification strategies that do not consider these factors increase the overall calcium intake, but risk supplying only non-absorbable calcium. In this light, the knowledge of the calcium amount in food matrices and of its available form, as well as of chemical changes occurring during digestion, are all critical factors to be considered to enable the best clinical management of calcium dietary intake and/or supplementation.

#### 2.1.2. Vitamin D Dietary Intake

Vitamin D is a pleiotropic hormone that regulates numerous biological functions, such as blood pressure, immunological responses and insulin production, and exerts beneficial effects in preventing cardiovascular diseases, diabetes, and certain types of cancer [23].

Vitamin D is principally known as a key and essential regulator of calcium homeostasis. Correct availability of vitamin D is, indeed, essential for maintaining healthy bone metabolism, by stimulating calcium and phosphate absorption from the gut and negatively controlling PTH secretion through the endocrine action of its active metabolite, calcitriol [24].

When we speak generically of vitamin D, we are commonly referring to a family of molecules sharing similarities in chemical structure and able to activate the endogenous vitamin D receptor. The two most common forms are vitamin D2 (or ergocalciferol) and vitamin D3 (or cholecalciferol). The former is of plant origin, while the latter is synthesized in humans and animals.

The main source of vitamin D3 in humans remains the skin, where 7-dehydrocholesterol (pro-vitamin D3) is transformed through a non-enzymatic reaction by ultraviolet radiation (UVB between 290 and 320 nanometers) into pre-vitamin D3. This then undergoes a temperature-dependent rearrangement, with the formation of vitamin D3. The minimum daily duration of sun exposure to achieve sufficient vitamin D production is difficult to establish as it is influenced by various parameters, such as latitude, season, time of day, age (for the same amount of sun exposure an older person produces 30% less), the surface area and thickness of skin exposed to the sun, the type of complexion and the use of sunscreen. Calcitriol synthesis is tightly regulated by PTH, calcemia and phosphoremia [25].

Vitamin D3 is also sourced from animal-derived foods, such as meat, egg yolk, dairy products, and some fatty fishes (salmon, trout and cod liver), while vitamin D2 is present in plants and mushrooms [25].

Vitamin D availability in the human body is mainly obtained from the skin synthesis (about 80%), while only the remaining 20% comes from the diet, either as vitamin D3 or vitamin D2. No single food or dietary regimen is capable alone of completely supplying the required amounts of vitamin D. As a consequence, given the importance of a correct apport of vitamin D for bone health, MEN1 patients should be advised to get adequate daily sun exposure during spring and summer, and/or take cholecalciferol supplements, to prevent deficiency/insufficiency of this important hormone (Table 1).

#### 2.1.3. Magnesium Dietary Intake

Magnesium is found in most whole foods, such as green leafy vegetables, legumes, and nuts. The recommended daily doses of magnesium are 240 mg for women and men (Table 1) [26].

Populations that consume more processed foods (cereals, sugars and more refined fats) have lower levels of magnesium, such as in the United States. A significant problem stems from the soil used for agriculture, which is becoming increasingly deficient in essential minerals, such as magnesium.

In several animal studies, dietary magnesium restriction promotes osteoporosis, bone fragility, micro-fractures of trabeculae and reduced mechanical properties of the bone. In contrast, in ovariectomized rats, magnesium supplementation increased osteocalcin, reduced PTH and deoxypyridinoline, and increased bone strength and fracture resistance. Also in humans, magnesium deficiency contributes to osteoporosis; in the Framingham study, magnesium intake was positively associated with bone mass density [27]. Recently, the protective effect of high magnesium intake on bone quality has been documented in healthy women using ultrasound measurement in the calcaneus [28]. Another study shows a positive effect of magnesium supplementation on bone mass accumulation in the hips of young women [29].

#### 2.1.4. Phosphorus Dietary Intake

Phosphorus is a primary element in the body and it is found almost exclusively in combination with oxygen in the form of phosphate. The estimated total phosphorus amount in a young adult is about 600 mg, 85% of which is found in the skeleton, as inorganic phosphorus, to form, together with calcium and magnesium, hydroxyapatite crystals. In addition, organic phosphorus is a major component of phosphoproteins, phospholipids and nucleic acids. In the blood, phosphorus is available in both the inorganic and the organic forms.

Phosphorus is taken in through the diet and eliminated through urine and sometimes feces. The regulation of calcium and phosphorus levels in the blood is dependent on each other and it is mediated by PTH, which, through calcitriol, increases calcium and phosphorus absorption by the intestine. In addition, if blood levels of phosphorus decrease, PTH and calcitriol can stimulate phosphorus release from the bones (through induction bone reabsorption) and, at the same time, increase the reabsorption of this element by the kidneys [30].

An adequate phosphorus intake is required for the mineralization of cartilage and osteoid tissue during endochondral ossification. A balanced diet provides sufficient amounts of phosphorus in most circumstances; a foodborne phosphorus deficiency is, thus, very unlikely. Phosphorus is found in high amounts in protein-containing foods, such as dairy products (i.e., milk: 1150 mg/L, Swiss cheese: 500 mg/100 g), meat, grains, beans, lentils, and nuts. The recommended dietary allowance is 1250 mg/day for children and adolescents during growth, and 700 mg/day for adults (Table 1). Under normal conditions, 60–70% of dietary phosphorus is absorbed [31,32].

#### 2.1.5. Vitamin K Dietary Intake

Vitamin K is a fat-soluble vitamin presenting two forms. The main type, named phylloquinone (vitamin K1), is present in green leafy vegetables like collard greens, kale, and spinach. The other type (vitamin K2) includes a series of menaquinones, which are present in modest amounts in some animal and fermented foods, and are predominantly produced by bacteria in the human body.

Vitamin K acts as a coenzyme for vitamin K-dependent carboxylase, an enzyme required for the synthesis of proteins involved in blood clotting, bone metabolism and other biological functions. Vitamin K can influence bone formation and effectively inhibit the reabsorption of bone mass [33]. Furthermore, there is a consistent line of evidence in epidemiological and human intervention studies demonstrating that vitamin K can not only increase BMD in osteoporotic individuals, but also reduce fragility fracture rate [34,35]. Specific studies are still needed to assess if vitamin K may be used in the prevention and treatment of osteopenia/osteoporosis. In general, a healthy diet, rich in fruit and vegetables, ensures an adequate vitamin K intake for the majority of the population.

#### 2.1.6. Potassium Dietary Intake

Potassium is a very important element for calcium homeostasis, since it increases urinary pH, making urine more alkaline and, thus, decreasing the risk of calcium oxalate crystal formation [36]. Therefore, a diet low in potassium content increases urinary calcium loss, while diets rich in potassium reduce calcium excretion. Potassium citrate is found in various vegetables, fruits, legumes and dairy products. Studies have shown a correlation between potassium intake and BMD [37,38]. Furthermore, increasing potassium citrate intake has been shown to improve the increased bone resorption observed in salt-rich diets. Higher potassium consumption, especially through fruit and vegetables, has been associated with higher initial BMD and less bone loss over time. The importance of ensuring adequate potassium intake from fruit and vegetables is a solid reason for promoting the recommendation to consume 5 to 10 servings per day.

#### 2.1.7. Sodium Dietary Intake

LARN 2014 recommends a sodium intake less than 1.5 g per day (Table 1).

Sodium increases calcium excretion, and higher calcium excretion is associated with lower BMD and increased risk of fractures. Consequently, it has been hypothesized that high sodium intake may be a direct risk factor for osteoporosis [39,40]. However, the regulation of sodium balance within the body is quite complex. An adequate sodium content throughout the body is necessary for the maintenance of central blood volume and renal perfusion and it is, therefore, tightly regulated by homeostatic defense mechanisms, mediated by the renin–angiotensin–aldosterone system (RAAS). The Korea National Health and Nutrition Examination Survey reported that osteoporosis was more frequently observed in postmenopausal women who consumed salt at ≥4001 mg per day, compared to those who consumed ≤2000 mg/day. A salt intake of ≥5001 mg was associated with a higher risk of osteoporosis in the femoral neck, with respect to a consumption of ≤2000 mg/day [41].

### 2.2. Dietary Interventions for Prevention and Management of Nephrolithiasis and Renal Function in MEN1 PHPT

PHPT in MEN1 is associated with a high risk of developing kidney stones, due mainly to the resulting hypercalciuria (a total urinary excretion of calcium in 24 h >250 mg in females or >300 mg in males) [42].

Calcium oxalate stones are the most common in MEN1 patients, but calcium phosphate stones can also form.

Since eating habits play an important role in the genesis of kidney stones, it is recommended to instruct MEN1 patients with dietary advice aimed at preventing/reducing major risk factors, such as those causing urine over-saturation for calcium oxalate and calcium phosphate (Table 2) [43].

#### 2.2.1. Drink Intake

One of the most effective strategies to reduce the risk of kidney stone formation is to increase daily fluid intake. It is estimated that for every additional 200 mL of fluids consumed per day, there is a 13% reduction in the risk of kidney stone formation. Daily fluid intake should therefore be high (>3 L on average) to obtain at least 2 L of urine per day, with personal adjustments depending on the type of work, climate, and lifestyle [44]. In addition to water, other types of beverages can also be consumed, but not all of them have the same protective effect. The intake of both lemon and orange juice, containing high citrate content and low fructose, has a protective effect that seems to be greater for orange juice, which in addition provides a higher alkaline load. This observation underlines the importance of the cation that accompanies the intake of citrate; if citrate is taken in the form of a proton, as in lemonade, it could neutralize the alkalizing effect of citrate [45].

Conversely, the intake of carbonated and sugary drinks, containing fructose, is directly associated with increased urinary excretion of calcium and oxalate, with a consequent increased risk of kidney stones, and this kind of beverage should be avoided by MEN1 patients.

#### 2.2.2. Calcium Dietary Intake

In hypercalciuric patients, dietary calcium intake should be at least 1200 mg per day (calcium should preferably be obtained from food, rather than supplements). As demonstrated in various studies, reducing dietary calcium intake is not an appropriate therapeutic strategy as this is associated with an increased risk of kidney stones. A randomized trial was conducted that compared a normal calcium diet (1200 mg per day), low in salt and animal proteins, with a low-calcium diet (400 mg per day) in a group of 120 men with recurrent calcium oxalate stones and hypercalciuria. The group that consumed normal amounts of calcium had a 50 per cent reduction in the risk of recurrence after 5 years compared to the group with a low-calcium diet [42,44]. In addition, urinary excretion of oxalate was higher in patients with low dietary calcium intake and lower in patients with high calcium intake, and low intakes of salt and animal proteins. This is because calcium in the gut acts as a chelator for several substances, including oxalate. With a low-calcium diet, there is an increase in the absorption of free oxalate, which increases oxaluria and over-saturation of calcium oxalate in the urine. Furthermore, balanced dietary calcium intake appears to have positive effects on kidney stone occurrence, regardless of its origin, whether from dairy or non-dairy sources [42,44,45].

#### 2.2.3. Sodium Dietary Intake

Dietary sodium levels should be limited to less than 1.5 g per day in adults, as recommended by the LARN 2014 [45]. Sodium chloride represents the main dietary source of sodium (sodium is 40% of the weight of sodium chloride). Although the daily intake of sodium required for proper homeostasis of the body is about 0.5 g per day, much higher doses are normally introduced. The average daily intake of sodium in the Italian population is calculated to be about 10.9 g in men and about 8.5 g in women; a single teaspoon of table salt contains about 2.3 g of sodium. Keeping sodium values within certain limits is difficult since it is added to any industrially processed food [45].

A study by Damasio et al., showed a direct association between a diet rich in sodium chloride and patients with idiopathic hypercalciuria and nephrolithiasis [46], compared to normocalciuric individuals. A high intake of sodium is, indeed, directly associated with urinary calcium excretion. Therefore, MEN1 patients, especially those with PHPT-derived hypercalciuria, should ideally limit their dietary sodium intake to less than 1.5 mg per day [45,46,47].

#### 2.2.4. Oxalate Dietary Intake

Calcium oxalate is a calcium salt of oxalic acid, a compound that occurs in needle-like crystals or raphides. In plants, these pointed needles, which are enclosed in vacuoles, are a defense mechanism against herbivores. Plants such as rhubarb, spinach, tea leaves, kiwi, beetroot, and green cabbage, but also nuts and chocolate, are rich in oxalate; thus, it is difficult to significantly limit its intake. This is why large quantities of oxalate are ingested every day and 50% of normal urinary excretion of oxalate is food-borne. The remaining amount is due to endogenous hepatic metabolism. This molecule has no nutritional function and is therefore eliminated through the urine. In the urine, oxalate binds calcium, increasing the over-saturation of calcium oxalate. Normally, intestinal absorption of oxalate is low and highly variable (approximately 10–15%). In individuals without malabsorption syndrome, intestinal oxalate absorption may only increase when intestinal ionized calcium is reduced, often due to high dietary consumption of phthalate (a calcium-binding molecule) and/or a low-calcium diet [45]. Currently, there are no agents available on the market that can reduce intestinal oxalate, but there is much attention on the use of enzymes derived from *Oxalobacter formigenes*, a microorganism capable of degrading intestinal oxalate. However, this therapeutic approach is still being studied. Therefore, the only way to reduce intestinal absorption of oxalate is to use adequate dietary sources of calcium or supplements in the case of oxalate-rich meals or try to limit consumption of the above-mentioned foods [42,45].

#### 2.2.5. Potassium Citrate Dietary Intake

Potassium citrate is the potassium salt of citric acid. It is found abundantly in some foods, such as citrus fruits (lemon and orange), and other types of fruit, such as kiwi and strawberries. Potassium citrate supplements can alkalize the urine to prevent the precipitation of calcium oxalate crystals. Potassium citrate treatment should be used to increase urinary citrate levels to >400 mg per day. The consumption of fruit and vegetables is therefore essential for MEN1 patients suffering from hypercalciuria, and it is useful to provide a sufficient amount of alkalis and citrates. Citrate also prevents the aggregation of calcium oxalate crystals that have already formed [46,47,48].

#### 2.2.6. Dietary Intake of Proteins of Animal and Vegetable Origin

Protein plays an important role in the formation of urinary stones. In one study, it was shown that in 18 hypercalciuric subjects with nephrolithiasis, a protein intake of 0.8 g/kg/day and 955 mg of calcium for 15 days significantly improved several urinary lithogenic risk factors. A significant decrease in both urinary calcium and oxalate and a simultaneous increase in urinary excretion of citrate was obtained [49]. In a study of three large cohorts, plant proteins were not associated with the risk of kidney stones, and higher potassium intake due to high vegetable intake was one of the most protective factors for nephrolithiasis. At the same time, milk protein intake was inversely associated with incident kidney stones. Only animal, non-dairy proteins appear to be harmful for the occurrence of kidney stones [45,48].

From all of the above, it is clear that behavioral and nutritional interventions are potentially beneficial and should be the first step in stone prevention.

## 3. Role of Nutrition Management in MEN1 Patients with GEP-NETs

MEN1 patients are prone to develop multiple GEP-NETs that originate in different sites of the gastrointestinal tract, although duodenum–pancreatic ones (DP-NETs) are the most frequent. The treatment of choice is almost always surgery with the ultimate aim of eradicating the primary tumor(s) [50,51].

At the clinical level, two groups of NETs can be distinguished, depending on whether or not there is an uncontrolled and excessive release of one or more hormones into the bloodstream by tumor cells, and whether or not a clinical syndrome related to the overproduction of tumor hormones appears, respectively, non-functioning tumors, not producing active substances, and functioning tumors, characterized by the increased production and secretion of biologically active hormones.

Functioning pancreatic–NETs (F-panNETs) are in turn subdivided, depending on the secreted hormone, into insulinoma, gastrinoma, somatostatinoma, tumors secreting vasoactive intestinal peptides (VIPomas), and tumors secreting glucagon [52].

The treatment of choice for panNETs and duodenal gastrinomas is, in the majority of cases, surgery, aimed at eradicating the primary tumor(s), preventing tumor progression, and control/cure endocrine syndromes due to hormone over-secretion.

Pancreatectomy, whether total or partial, consists of the removal not only of the pancreas (total or partial) but also of the first part of the intestine (duodenum and part of the jejunum), the spleen and the gallbladder. Sometimes, the resection may also extend to part of the stomach. The removal and/or resection of all these components leads to alterations in the mechanisms of digestion and absorption. This is due to the reduction in or complete lack of digestive enzymes, such as protease, lipase and amylase, normally produced by the exocrine pancreas, which are responsible for breaking down ingested food to be absorbed in the intestine. All this entails a high risk of malnutrition. Therefore, it is very important to make the operated MEN1 patients aware of their individual post-operative new digestive functions, and be instructed on nutritional indications to be followed to reduce the symptoms of pancreatic resection and to face the functional recovery phase in the best possible way. Commonly, patients need to be made protagonists of their course of post-operative treatment, in order to not suffer the nutritional indications as an imposition, but as a necessary change to have as a best quality of life as possible.

Currently, nutritional and vitamin status is a neglected area for MEN1 patients with DP-NETs. There are clinical practice guidelines and guidelines for diagnosis, treatment and medical management, but little for nutritional management and support [53].

### 3.1. Prevention of Post-Surgical Malnutrition in MEN1 Patients Operated for GEP-NETs

In MEN1 patients, pancreatic resections lead in most cases to alterations in the mechanisms of nutrient digestion and absorption. Poor digestion leads to a worsening of the patient’s general quality of life, who experiences a whole series of gastrointestinal manifestations, such as bloating, flatulence, abdominal cramps and steatorrhea. In addition, operated patients present with various systemic manifestations due to an overall nutrient deficiency, such as anemia due to reduced absorption of iron, folate and vitamin B12, ascites due to reduced protein absorption, or even neuropathy due to a lack of B vitamins.

Lifelong replacement therapy with digestive enzymes derived from the porcine exocrine pancreas is the intervention of choice for the management of malabsorption due to exocrine pancreas insufficiency [50,51].

Due to the high probability of malnutrition, MEN1 patients should be advised to undergo periodic screening to define the level of risk of nutritional deficiency. This should be part of the routine care of every MEN1 patient, as malnutrition has substantial negative consequences, including increased mortality, poorer quality of life and increased healthcare costs. Malnutrition is often associated with cachexia, a condition characterized by a loss of fat and muscle mass. To counteract the onset of malnutrition, it is very important to periodically monitor certain parameters, including body weight, appetite and regular meal intake.

In general, the diet of MEN1 patients with DP-NET who are about to undergo surgical resection of the pancreas or who have already undergone surgery should include an adequate amount of all nutrients, such as proteins, fats, carbohydrates, vitamins and minerals. Sometimes, in addition to meals, it is also important to provide targeted nutrient supplements [54].

A balanced and correct diet should prevent the loss of body mass, promote tissue recovery/maintenance and improve the patient’s quality of life and ability to perform daily and work activities.

According to the most recent published studies, the diet of a pancreatectomy patient should be normoproteic and normocaloric, with a controlled intake of lipids and a reduced fiber content. In detail, the following basic requirements, based on reference weight, should be estimated:-Calorie intake: 25–30 kcal/kg-Protein intake: 0.85–1 g/kg/day-Fat intake: no more than 30% (approx. 30–40 g/day)-Carbohydrate intake: 45–50% (with fast-absorbing sugars < 10% of total Kcal)-Fiber intake: <10 g/day (to be assessed on a case-by-case basis according to the frequency/type of bowel movements and the presence of postoperative diabetes mellitus) [53].

Regarding micronutrients (iron, magnesium, zinc and calcium), fat-soluble vitamins (A, D, E) and vitamin B12, periodic monitoring using specific blood tests is necessary.

In the post-surgery period, it is necessary to adapt the diet to the new functional capacities of the gastrointestinal tract and to prevent certain symptoms that may arise in the early postoperative period, such as nausea, vomiting, loss of appetite and in particular diarrhea/steatorrhea. To this end, the resumption of oral nutrition with a low-fat and low-fiber diet is indicated [54].

### 3.2. Management of Endocrine Insufficiency

After pancreatic surgery, endocrine insufficiency may also occur due to a reduction in insulin production by pancreatic β-cells, resulting in pancreatogenic diabetes (type IIIC diabetes).

The therapeutic target was to achieve glycated hemoglobin < 7% and a fasting blood glucose between 70 and 130 mg/dL [55].

Also in this case, patients must be recommended following a healthy diet and some precautions, necessary to maintain the correct post-surgery blood sugar levels, such as limiting the intake of carbohydrates, reducing simple sugars, which are rapidly absorbed, and limiting foods with a high glycemic index and load [6,56].

### 3.3. Dietary Indications in MEN1 Patients for the Management of Post-Surgical Symptoms

Surgery entails various consequences on the functioning of the digestive mechanisms that may occur differently depending on the patient and not necessarily in all persons undergoing the procedure. Symptoms such as nausea, vomiting, diarrhea, steatorrhea and early satiety can occur, which if not promptly managed can cause dehydration and weight loss, worsening patient quality of life and lengthening the time of recovery after surgery. Correct management of these symptoms is very important to maintain proper body weight and speed up recovery.

### 3.4. Nutritional Recommendations in MEN1 Patients Treated with Somatostatin Analogs

Somatostatin analogues (SSAs), octreotide and lanreotide, were shown to be effective in MEN1, to both reduce excessive hormone secretion, controlling NET-derived endocrine syndromes, and to control tumor growth, reducing tumor size, and are, thus, commonly used in MEN1 patients with GEP-NETs not suitable for surgery, those who refused the surgical intervention, or to shrink tumor before surgery [51].

However, in addition their beneficial effects, SSAs negatively affect the digestive system, slow down the blood flow of visceral vessels and the motility of intestinal transport, and, above all, inhibit the secretion of pancreatic and intestinal hormones, all resulting in alterations in nutrient dietary intake. Therefore, patients receiving this type of therapy should be carefully and constantly monitored by a nutrition expert, to compensate, as much as possible, for the adverse effects of the therapy on nutrient bioavailability and intestinal absorption. In particular, patients should have their blood glucose values checked in the days immediately following SSA administration. Indeed, a reduction in insulin synthesis can occur as a consequence of the drug action, resulting in post-prandial glycemic peak. It is, thus, strongly indicated not to eat meals with a high sugar content. Processed foods with a high fat content should also be avoided to facilitate digestion and avoid steatorrhea.

Currently, there are no national dietary guidelines developed specifically for NETs [50,54,57], especially in patients under treatment with SSAs.

## 4. Conclusions

In conclusion, it is essential to underline the importance of managing MEN1 patients through a multidisciplinary approach, which should include also the figure of the nutritionist/dietitian, a specific aspect that is currently missing in the global management of MEN1 syndrome.

Numerous studies have shown that a balanced diet and an active lifestyle can help in prolonging and improving the quality of life of the general population. This is even more relevant in patients with MEN1 syndrome.

Healthy eating and an active lifestyle represent the only modifiable factors that can be implemented to promote personal well-being and ameliorate the quality of life in patients with genetic-caused tumor(s), such as MEN1 patients and *MEN1* mutation carriers.

The nutritionist/dietitian should develop a personalized nutrition plan, considering individual characteristics of patient, such as age, sex, personal history, and preferences.

The Mediterranean diet, with its richness in fruits, vegetables, citrus fruits, cruciferous vegetables and the integration of phytotherapeutic compounds, represents an appropriate nutritional choice, with particular attention to the daily intake of calcium, a crucial element in food planning for MEN1 patients.

## Figures and Tables

**Table 1 nutrients-16-01576-t001:** Recommended and adequate daily intakes of specific nutrients for bone health.

Patient Category	Age Range (Gender)	Ca (mg)	VIT D (µg)	Mg (mg)	P (mg)	VIT K (µg)	K (g)	Na (g)
Newborn	6–12 months	*260*	*-*	*80*	*275*	*10*	*0.7*	*0.4*
	1–3 years	**700**	**15**	**80**	**460**	**50**	**1.7**	**0.7**
Children	4–6 years	**900**	**15**	**100**	**500**	**65**	**2.4**	**0.9**
	7–10 years	**1100**	**15**	**150**	**875**	**90**	**3.0**	**1.1**
	11–14 (Males)	**1300**	**15**	**240**	**1250**		**3.9**	**1.5**
	11–14 (Females)	**1300**	**15**	**240**	**1250**		**3.9**	**1.5**
Adults	19–50 years	**1000**	**15**	**240**	**700**	**140**	**3.9**	**1.5**
	51–70 years (Males)	**1000**	**15**	**240**	**700**	**170**	**3.9**	**1.5**
	51–70 years (Females)	**1200**	**15**	**240**	**700**	**170**	**3.9**	**1.5**
	>75 years	**1200**	**20**	**240**	**700**	**170**	**3.9**	**1.2**
Women during pregnancy/breastfeeding	14–18 years	**1200**	**15**	**240**	**700**	**140**	**3.9**	**1.5**
	19–50 years	**1000**	**15**	**240**	**700**	**140**	**3.9**	**1.5**

Population recommended intakes (PRI) are reported in bold, while adequate intakes are reported in italics. When gender is not specified, it includes both males and females.

**Table 2 nutrients-16-01576-t002:** Recommendations for patients to reduce the risk of developing kidney stones.

Factor Influencing Development of Kidney Stones	Recommendations for Patients
Quantity and type of drink(s)	- Increase fluid intake: over 3 L/day (Water. orange and lemon juice) - Decrease consumption of fructose-containing drinks
Calcium intake	- Check that dietary calcium intake reflects the Population Recommended Intake (PRI) values for age and gender
Sodium intake	- Reduce dietary sodium intake to less than 1.5 g/day
Oxalate intake	- Limit the use of foods rich in oxalates - Increase the daily calcium intake, especially if the meal has a high oxalate content
Potassium intake	- Increase the intake of potassium-rich foods, mentioned to be at least 400 mg/day
Protein intake	- Favor the intake of proteins of vegetable origin rather than those of animal origin

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
