# Peer review of "Role of Nutrition in the Management of Patients with Multiple Endocrine Neoplasia Type 1"

_nutrients, 2024, doi:10.3390/nu16111576_

Round 1

Reviewer 1 Report

Comments and Suggestions for Authors

Many nutrients contribute to the prevention of bone mineral loss. Therefore, it is recommended that authors could discuss the role of other nutrients such as potassium, sodium, and vitamin K in this review article. The author also needs to  discuss how much calcium coming from drink intake in comparison with dairy foods.

Comments on the Quality of English Language

Minor editing of English language is required.

Reviewer 2 Report

Comments and Suggestions for Authors

This is an interesting article by M. Marinariet al.

I would like to address a small number of suggestions to you.

Please correct all references according to journal stile.

References should be described as follows:

Journal Articles:

1.      Author 1; Author 2. Title of the article. Abbreviated Journal Name Year; Volume: page range.

Introduction

Page 2 line 48

Please write citations as [1-9].

Page 3 lines 107-115

The first paragraph of the section 2.1 is better to presents in introduction.

Please rewrite the paragraph "MEN1 is characterized by a high prevalence of PHPT, occurring at a very early age compared to its sporadic counterpart and progressively increasing with age up to over 95% of cases after 50 years of age. MEN1 PHPT contributes to notably affecting bone health; the most common skeletal alteration is the early-onset reduction in bone density. Some studies showed that MEN1 patients with PHPT manifest a significant reduction in bone mineral density (BMD), associated with an increased likelihood of fragility fractures, compared to the general population.[12], [13], [14], [15]. Recent studies have stated that 113 this depends not only on the increased secretion of PTH and its consequent effect on bone resorption but that the presence of the inactivating mutation in the MEN1 gene can directly affect the metabolism of osteoid line cells [16]."

Moreover, when you write "Recent studies" you must refer at least 2 references.

Page 4

Please add reference for article Bacciottini et al., which you analyzed. 

Moreover, at the end of this paragraph, please delete the reference 18, because the article that you describe, by R. P. Heaney and M. S. Dowell, is reference 20.

Page 4 and 5

The sections "2.1.2 Vitamin D dietary intake" and "2.1.4 Phosphate dietary intake" are too short and do not reveal the extent of the problem.

Page 6

Paragraph "2.2.3 Sodium dietary intake". The role of sodium intake in calcium metabolism must be better specified. Please add and analyzed an article by Damasio, P.C et al. The role of salt abuse on risk for hypercalciuria. Nutr. J. 2011, 10, 3.

Page 9

Paragraph "3.4 Nutritional recommendations in MEN1 patients treated with somatostatin analogs."

All recommendations in this paragraph are chaotic written.

Please rewrite this paragraph

In my opinion, it is necessary to add some tables and schematic figures to the review article.

Round 2

Reviewer 2 Report

Comments and Suggestions for Authors

All my corrections and recommendations were added.